# Enhancing the Anti-Leukemic Potential of Thymoquinone/Sulfobutylether-β-cyclodextrin (SBE-β-CD) Inclusion Complexes

**DOI:** 10.3390/biomedicines11071891

**Published:** 2023-07-04

**Authors:** Eltayeb E. M. Eid, Salah Abdalrazak Alshehade, Amer A. Almaiman, Sareh Kamran, Vannajan Sanghiran Lee, Mohammed Abdullah Alshawsh

**Affiliations:** 1Department of Pharmaceutical Chemistry and Pharmacognosy, Unaizah College of Pharmacy, Qassim University, Unaizah 51911, Saudi Arabia; 2Department of Pharmacology, Faculty of Medicine, Universiti Malaya, Kuala Lumpur 50603, Malaysia; 3Unit of Scientific Research, Applied College, Qassim University, Unaizah 51911, Saudi Arabia; 4Department of Chemistry, Faculty of Science, Universiti Malaya, Kuala Lumpur 50603, Malaysia

**Keywords:** thymoquinone, SBE-β-CD, leukemia, drug delivery system, telomerase inhibition, apoptosis

## Abstract

Leukemia, a condition characterized by the abnormal proliferation of blood cells, poses significant challenges in cancer treatment. Thymoquinone (TQ), a bioactive compound derived from black seed, has demonstrated anticancer properties, including telomerase inhibition and the induction of apoptosis. However, TQ’s poor solubility and limited bioavailability hinder its clinical application. This study explored the use of Sulfobutylether-β-cyclodextrin (SBE-β-CD), a cyclodextrin derivative, to enhance the solubility and stability of TQ for leukemia treatment. SBE-β-CD offers low hemolytic activity and has been successfully employed in controlled drug release systems. The study investigated the formation of inclusion complexes between TQ and SBE-β-CD and evaluated their effects on leukemia cell growth and telomerase activity. The results indicated that the TQ/SBE-β-CD complex exhibited improved solubility and enhanced cytotoxic effects against K-562 leukemia cells compared to TQ alone, suggesting the potential of SBE-β-CD as a drug delivery system for TQ. The annexin V-FITC assay demonstrated increased apoptosis, while the qPCR quantification assay revealed reduced telomerase activity in leukemia cells treated with TQ/SBE-β-CD, supporting its anti-leukemic potential. The molecular docking analysis indicated a strong binding affinity between TQ and telomerase. However, further research is needed to optimize the apoptotic effects and minimize necrosis induction. In conclusion, TQ/SBE-β-CD shows promise as a novel strategy for leukemia treatment by inhibiting telomerase and enhancing the cytotoxic effects of TQ, offering a potential solution to overcome the limitations of TQ’s poor solubility and bioavailability.

## 1. Introduction

The medicinal value of thymoquinone, a phytochemical compound found in the oil of *Nigella sativa* seeds, otherwise known as black seed, has been recognized across various cultures worldwide, from southwestern Asia to the Mediterranean and Africa. Black seed’s volatile oil, rich in thymoquinone, presents a diverse range of pharmacological effects, including antioxidant, anti-inflammatory, and anticancer properties. Consequently, it has the potential to prevent and treat numerous health conditions [1,2].

These effects of thymoquinone are tied to its capacity to neutralize free radicals and reactive oxygen species, which are central to the development of many diseases. It is therefore proposed as a protective agent against diseases related to oxidative stress such as cardiovascular disease and cancer. Moreover, the compound’s strong anti-inflammatory effects limit the production of pro-inflammatory mediators, suggesting potential therapeutic value for inflammatory diseases like rheumatoid arthritis and inflammatory bowel disease. Thymoquinone also exhibits antihistaminic, hypoglycemic, hypolipidemic, and antimicrobial effects; thus, it is of interest for a wide range of health conditions [3,4].

The anticancer properties of thymoquinone have been particularly highlighted in various preclinical trials. It has proven capable of inhibiting the proliferation of different cancer cell lines and triggering apoptosis—the programmed cell death crucial for curbing cancer proliferation. Moreover, its antitumor effects are multifaceted, modulating molecular pathways involved in cell proliferation, apoptosis, and metastasis, such as p53, p38, and STAT3. The compound has shown potential against different cancer types, including breast, colorectal, pancreatic, lung, and prostate cancer, where it inhibits tumor growth, angiogenesis, and metastasis [5,6,7]. However, to corroborate these promising preclinical results, clinical trials in humans are needed to confirm the safety, efficacy, and optimal dosage of thymoquinone.

In acute myeloid leukemia research, thymoquinone has shown potential to induce the re-expression of tumor suppressor genes (TSGs) previously silenced by epigenetic mechanisms like DNA methylation. Leukemia cells treated with thymoquinone showed a significant decrease in growth and an increase in apoptosis. This effect is linked to thymoquinone’s ability to bind the active pockets of JAK2, STAT3, and STAT5, thus inhibiting their enzymatic activity. Additionally, thymoquinone has been seen to enhance the re-expression of SHP-1 and SOCS-3, two TSGs, via demethylation. Such findings highlight thymoquinone’s potential as a therapeutic agent in the treatment of acute myeloid leukemia [8].

On a molecular level, telomerase plays a key role in the regulation of cancer cells. This ribonucleoprotein enzyme is crucial in maintaining the ends of chromosomes by adding telomeric repeats to these ends. Interestingly, telomerase activity is found in almost 90% of tumor cells, including leukemia cells, but is typically absent in adjacent normal cells [9]. Despite the apparent contradiction, given the link between telomerase deficiency and leukemia onset, it has been proposed that the gradual loss of telomeric DNA, or telomere attrition, fosters genomic instability and predisposes cells to tumor development. Therefore, to sustain stable and continuous proliferation, a tumor cell must reactivate telomerase, making this enzyme a promising target for cancer therapy [10,11].

While prior studies have shed light on the effects of thymoquinone on tumor suppressor genes, particularly in the context of acute myeloid leukemia, the literature still has significant gaps regarding the mechanistic actions and the role of this compound in relation to telomerase activity in cancer cells. The current study aimed to bridge this gap by exploring the potential interplay between thymoquinone encapsulated in sulfobutylether-β-cyclodextrin and telomerase in leukemic cells. The unique contribution of our research is the comprehensive investigation of thymoquinone’s ability to modulate telomerase activity, an aspect not fully explored in previous studies. Thus, we aimed to enhance our understanding of thymoquinone’s anticancer properties and so provide novel insights that may be instrumental in developing more effective therapeutic strategies against leukemia.

## 2. Materials and Methods

### 2.1. Preparation of the Inclusion Complex 

We applied a lyophilization technique to formulate the inclusion complex of thymoquinone with sulfobutylether-β-cyclodextrin (SBE-β-CD), replicating the process detailed in our prior research [12,13]. We mixed a balanced 1:1 molar combination of TQ and SBE-β-CD, dissolving it in 20 mL of ultrapure water. The optimized 1:1 molar ratio between TQ and SBE-β-CD was well characterized in solution, and a thermodynamically stable complex with validated binding energy values. Therefore, the inclusion complex of TQ and SBE-β-CD will be used in our future in vivo studies for the further improvement of TQ’s pharmacokinetic profile. This solution was subsequently agitated in a rotary shaker at ambient temperature for an extended period of 72 h before being strained through a 0.45 μm filter. The resultant translucent solution was flash-frozen at an extremely low temperature of −80 °C, after which it underwent a lyophilization process wherein 10 mL of the inclusion complex solution of TQ/SBE-β-CD was completely frozen using a −80 °C fridge in a 40 mL ambered freeze dryer flask for 48 h; then, the frozen sample was fit into a loop of a freeze-dryer head (Labconco Corporation, Kansas City, MO, USA) for 72 h at −85 °C with a vacuum of 0.019 mbar, followed by a final drying stage for up to 4 h at 76 ◦C with a vacuum of 0.010 mbar. Then, the lyophilized powder was stored in an airtight container protected from light to be used in different experiments.

### 2.2. Cytotoxicity

K-562 leukemia cells (ATCC, Manassas, VA, USA, Cat. CCL-243) were cultured in Iscove’s Modified Dulbecco’s Medium (IMDM) (ATCC, USA Cat. 30-2005), supplemented with 10% fetal bovine serum (FBS) and a 1% solution of 10,000 I.U./mL penicillin and 10,000 μg/mL streptomycin (ATCC, USA, Cat. 30-2300). For each time point, a separate plate was utilized. Five wells of a tissue culture microplate (96 wells) with a clear bottom were populated with 10,000 K-562 cells each, containing 100 µL of medium per well. Following this, cells were treated with various concentrations of sulfobutylether-β-cyclodextrin (SBE-β-CDs, ranging from 5.72 to 0.04 mg/mL); thymoquinone (TQ, ranging from 100 to 0.78 μg/mL); or a complex of TQ/SBE-β-CDs (ranging from 100 to 0.78 μg/mL). One set of cells served as the negative control and was left untreated. Post-treatment, the cells were incubated at 37 °C in a 5% CO2 environment for varying durations: 24, 48, 72, and 96 h. After each incubation period, Cell Counting Kit 8 (WST-8/CCK8) solution was directly added to the test wells containing the incubated cells and was left for two hours. Following this, the absorbance was measured at 450 nm using a VersaMax reader (Molecular Devices, LLC, San Jose, CA, USA). The cell toxicity percentage was calculated using Equation (1) [14].
(1)Cytotoxicity%=Negative controlAbs−SampleAbsNegative controlAbs×100

### 2.3. Caspase-Glo^®^ 3/7, 8, and 9 Assays

The activity of Caspase-Glo^®^ 3/7, 8, and 9 enzymes was assessed in a 96-well plate using Caspase-Glo^®^ 3/7 Assay, Caspase-Glo^®^ 8 Assay, and Caspase-Glo^®^ 9 Assay kits (Promega, Madison, WI, USA, Cat. G8091, Cat. G8201, and Cat. G8211, respectively), as per the guidelines provided by the manufacturer. Briefly, after a 96-hour incubation period of the K-562 cells with SBE-β-CDs (at concentrations of 0.36, 0.72, and 1.43 mg/mL); TQ (at concentrations of 6.25, 12.5, and 25 µg/mL); or a TQ /SBE-β-CDs complex (at concentrations of 6.25, 12.5, and 25 µg/mL), an analysis was carried out. The negative control comprised cells treated with vehicle in a medium. Each type of caspase was examined in a separate set of wells. For each well on a white-walled plate that contained 100 µL of either negative control cells or treated cells in a culture medium, 100 µL of Caspase-Glo Reagent was added. Subsequently, the contents of the wells were mixed gently using a plate shaker at a speed of 300–500 rpm for half a minute, followed by an incubation period of an hour at room temperature. The luminescence for each sample was then measured using a plate-reading luminometer.

### 2.4. Annexin-V/PI Flow-Cytometry Protocol

K-562 cells, at a density of 1 × 10^6^ cells, were distributed across a T25 culture flask, with three identical flasks for control purposes (these were unstained, stained only with annexin, and stained only with propidium iodide (PI)). After a 96-hour incubation period with various treatments, such as SBE-β-CDs (at 0.72 mg/mL), TQ (at 12.5 µg/mL), or a TQ/SBE-β-CDs complex (at 12.5 µg/mL), the cells from each T25 flask were collected. These cells were then washed in 2 mL 1× PBS (without calcium or magnesium). Following this, the cells were resuspended in 1 mL of 1× Annexin V binding buffer. Once centrifuged and with the supernatant decanted, the cells were resuspended in 100 μL of 1× Annexin V binding buffer with the addition of 5 μL (or 1 µg/mL) of Annexin V Alexa Fluor 488. After incubating for 15 min, 100 μL of 1× Annexin V binding buffer was added along with 4 μL of PI (at 100 µg/mL), and this was then left in the dark at room temperature for a further 15 min. Finally, the cells were processed through a flow cytometer (specifically, a BD FACSCanto II), analyzing a total of 10,000 events.

### 2.5. Telomerase Activity Quantification qPCR Assay (TAQ) 

The Telomerase Activity Quantification qPCR Assay (TAQ) kit (ScienCell Research Laboratories, Carlsbad, CA, USA, Catalog #8928) was employed as per the guidelines provided by the manufacturer. In brief, between 2 and 5 million cells were collected and washed using PBS. During the entirety of the experiment, the cell pellets, cell lysates, and reagents were kept on ice. To every milliliter of cell lysis buffer, 1 μL of 0.1M PMSF and 0.3 μL of 14.3M β-mercaptoethanol were added before utilization. Following cell lysis, the samples were centrifuged at 12,000× *g* for a span of 20 min at a temperature of 4 °C. Subsequently, 15 μL of supernatant was moved to a new, pre-chilled tube. The telomerase reaction was prepared by combining 0.5 μL of the cell lysate sample or cell lysis buffer (which acted as a negative control), 4 μL of telomerase reaction buffer, and 15.5 μL of nuclease-free H_2_O. The reaction samples were incubated at a temperature of 37 °C for 3 h, after which the reactions were halted by heating the samples at 85 °C for 10 min. Lastly, qPCR reactions were implemented by combining 1 μL of the post-telomerase reaction sample, 2 μL of primer, 10 μL of GoldNStart TaqGreen qPCR Mastermix, and 7 μL of nuclease-free H_2_O.

The qPCR program was subsequently set up in accordance with the manufacturer’s instructions. The concluding phase encompassed performing qPCR to analyze telomere production by telomerase. The quantification cycle (ΔCq), a pivotal concept in qPCR analysis, refers to the difference in the cycle threshold (Cq) values of the untreated and treated cells. The Cq value refers to the number of cycles required for the fluorescent signal to cross the threshold in qPCR, and lower Cq values generally imply a higher initial target concentration. The ΔCq was calculated as the Cq of the untreated cells minus the Cq of the treated cells as per Equation (2). The relative telomerase reduction activity, a measure of how much the telomerase activity decreased due to treatment compared to the untreated cells, was calculated using Equation (3). This calculation leveraged the power of 2 to the negative ΔCq, thereby transforming the logarithmic scale of the ΔCq value to a linear scale that could be understood more intuitively. This allowed for comparisons of the relative telomerase activity between different samples.
ΔCq = Cq_(Untreated cells)_ − Cq_(Treated cells)_(2)
Relative telomerase reduction activity = 2^−ΔCq^(3)

### 2.6. Molecular Docking

AutoDock VINA (V. 1.1.2) was used to carry out molecular docking studies [15], aimed at understanding the potential interactions between SBE-ß-CDs and TQ, as well as telomerase and TQ. The 3D structures for SBE-ß-CDs, TQ, and telomerase were procured from PubChem and the Protein Data Bank (PDB) (ID: 2BCK), respectively. To these structures, hydrogens were added, non-polar hydrogens were merged, and Gasteiger charges were computed using AutoDockTools. During docking, the Lamarckian genetic algorithm was used as the search algorithm, which is particularly effective for high-dimensionality and multi-modal problems. The grid box was adjusted to encompass the entire protein, and an empirical scoring function was used, which considered both the steric and electrostatic terms. Constraints were not applied during the docking procedure, allowing a complete exploration of the conformational space. An exhaustiveness level of 100 was set to ensure a comprehensive exploration of the conformational space. After docking, the resulting complexes were ranked based on their predicted binding affinities, which were calculated using the scoring function of AutoDock VINA. The conformation with the most favorable binding energy was selected for further analysis. To scrutinize the interaction profiles, including hydrogen bonds and hydrophobic interactions between TQ and the proteins, Discovery Studio Visualizer was utilized. The docking procedure produced a total of 50 conformations, and the one with the lowest energy was chosen for each complex. This comprehensive analysis provided valuable insights into the interactions between TQ and the proteins.

### 2.7. Molecular Dynamic Simulation 

To gain a comprehensive understanding of the dynamic behavior of the telomerase–TQ complex and offer valuable insights into their interaction at the molecular level, a molecular dynamics simulation was carried out using Nanoscale Molecular Dynamics (NAMD) software (Version, 2.14). The 3D structure of the complex was acquired from the Protein Data Bank (PDB). Any missing residues or atoms were appended using the Swiss PDB viewer, with care taken to assign the protonation states of the ionizable residues at a physiological pH. The potential energy function for the simulation was defined using the CHARMM (Chemistry at HARvard Macromolecular Mechanics) force field, and the parameters specific to TQ were generated via the SwissParam tool. The complex was then solvated in a TIP3P water-molecule-populated cubic box, which extended 20 Å beyond the complex in every direction. Counter ions were subsequently incorporated to neutralize the system, and a physiological salt concentration was achieved by adding NaCl. An energy minimization step was executed using the conjugate gradient method to alleviate steric clashes and relax the system. Following this, the system was incrementally heated to 37 °C over a 100 ps period and equilibrated under constant temperature and pressure (NPT ensemble) for 1 ns. The production run was then initiated post-equilibration and lasted for 100 ns, a duration justified by the need for the comprehensive sampling of the dynamic interaction. Trajectories were saved every 10 ps for later analysis. To track the system’s stability, the root mean square deviation (RMSD) of the protein backbone atoms was monitored. Binding interactions were examined using the LIGPLOT program and visualized with VMD (Visual Molecular Dynamics), with an integration time step of 2 fs applied to ensure accurate results. This approach thus offered a detailed exploration of the molecular-level interplay between telomerase and TQ.

### 2.8. Statistical Analysis

Statistical analyses were carried out using GraphPad Prism software (version 8.4). A one-way analysis of variance (ANOVA) was employed, followed by a post hoc Dunnett’s multiple comparisons test to examine the differences between group means. Statistical significance was determined as a *p*-value less than 0.05. All experiments were performed in triplicate to ensure reliable and reproducible results. The results are presented as the mean ± standard deviation. Any *p*-value less than 0.05 was considered to indicate a statistically significant difference.

## 3. Results

### 3.1. Cytotoxicity

The cytotoxic effects of thymoquinone (TQ) and thymoquinone encapsulated with SBE-ß-cyclodextrins (TQ/SBE-ß-CDs) were examined using a Cell Counting Kit-8 (WST-8/CCK8). As depicted in Figure 1, the percentage of cell inhibition reached a plateau after 24, 48, and 72 h, remaining below 50%. However, after 96 h of treatment, the maximum inhibition effects of TQ and TQ/SBE-ß-CDs were found to be 69.7 ± 5.72 and 89.1 ± 1.31, respectively. There was a significant difference in the inhibition effects of TQ and TQ/SBE-ß-CDs at a concentration of 25µg/mL after 96 h (*p* < 0.05). Using a non-linear model (variable slope—four parameters), the half-maximal inhibitory concentrations (IC50) for TQ and TQ/SBE-ß-CDs were determined to be 8.5 µg/mL and 9.9 µg/mL, respectively, after 96 h of treatment.

### 3.2. Annexin V-FITC Assay

An annexin V-FITC assay was used to evaluate the induction of cellular apoptosis by thymoquinone (TQ) alone and TQ encapsulated with SBE-ß-cyclodextrins. The flow cytometry data, presented in Figure 2, indicated that the percentage of viable cells was highest in the group treated with SBE-ß-CDs alone, with 95.8% alive and an insignificant percentage of apoptotic, dead, and necrotic cells. Conversely, the group treated with TQ showed the highest percentage of early apoptotic cells, at 18.8%. When TQ was encapsulated with SBE-ß-CDs and administered to the cells, the percentage of dead cells (late apoptosis) and necrotic cells rose from 17.8% to 23.8% and from 10.8% to 26.5%, respectively. Figure 2 also includes a bar chart to facilitate the comparison of the different treatment outcomes.

### 3.3. Effect of TQ and TQ/SBE-ß-CDs Therapy on Caspase Pathways

To delve deeper into the effects of the treatments on leukemia cells, the bioluminescent intensity of caspase 3/7, caspase 8, and caspase 9 was measured after 96 h of treatment. Figure 3 reveals that treatment with TQ at a dosage of 25 µg/mL primarily activated caspase 9, followed by caspase 3/7. When TQ was encapsulated with SBE-ß-CDs and applied to the cells, it exclusively activated caspase 8 at a dose of 25 µg/mL. Moreover, it triggered caspase 9 at doses of 6.25 µg/mL, with more pronounced activation at 12.525 µg/mL and 25 µg/mL. These findings suggested that the combined treatment operated via both caspase 8 and caspase 9 pathways, with a more substantial activation of the mitochondrial intrinsic pathway. 

### 3.4. Telomerase Activity Quantification qPCR Assay

The telomerase activity was quantitatively evaluated among thymoquinone (TQ), SBE-ß-cyclodextrins (SBE-ß-CDs), and TQ/SBE-ß-CDs, as presented in Figure 4. The telomerase activity of SBE-ß-CDs was marginally lower than that of the control, with a decrease of 1.406 ± 0.35. In contrast, the telomerase activity of TQ and TQ/SBE-ß-CDs was significantly reduced, with respective reductions of 18.70 ± 2.199 fold and 43.46 ± 5.568 fold compared to the negative control (Figure 4).

### 3.5. Molecular Docking Results

Molecular docking studies using AutoDock VINA 1.1.2 were conducted to examine the potential interactions of two complexes: SBE-ß-cyclodextrins (SBE-ß-CDs) with thymoquinone (TQ) and telomerase (ID: 2BCK) with TQ. The top-scoring pose demonstrated an affinity of −4.1 kcal/mol, suggesting a likelihood of SBE-ß-CDs binding with TQ. Additionally, the nature and location of the interactions between SBE-ß-CDs and TQ were identified (as shown in Figure 5A–D). A conventional hydrogen bond with a distance of 2.86 Å was identified between H35 (hydrogen donor) and O (hydrogen acceptor) (Table 1). A second conventional hydrogen bond was found with a distance of 2.21 Å, between H77 (hydrogen donor) and O (hydrogen acceptor). These hydrogen bonds significantly contributed to the stability of the SBE-ß-CDs and TQ complex.

The molecular docking technique was further utilized to examine potential interactions between thymoquinone (TQ) and telomerase (chains A and B). The findings indicated that TQ interacted with chain A of telomerase, with the top-scoring pose (mode 1) demonstrating an affinity of −6.65 kcal/mol, equivalent to an inhibition constant (Ki) of 13.45 μmol. This reflected a strong binding affinity between telomerase and TQ. The subsequent top-scoring poses (modes 2 and 3) revealed affinities of −5.93 and −5.76 kcal/mol, respectively (Figure 6A–D). Both poses exhibited low values for the “lower bound of the RMSD” (1.97 and 1.55 Å, respectively) and the “upper bound of the RMSD” (3.08 and 3.81 Å, respectively) relative to the ligand’s best mode (i.e., mode 1). This suggested the relative stability of these interactions. Key interactions between telomerase and TQ were observed and are detailed in Table 2. Notably, a carbon hydrogen bond was identified, with a distance of 2.75 Å between the TQ hydrogen donor and the telomerase chain A THR143 hydrogen acceptor. Additionally, a pi-donor hydrogen bond was seen with a distance of 2.95 Å between the TQ hydrogen donor and chain A’s TYR116 pi-orbitals. Several alkyl interactions were also identified, as detailed in Table 2.

### 3.6. Molecular Dynamic Simulation Results

To measure the extent of the structural transformations in both the protein and ligand of the docked complexes, we initially computed the root mean square deviation (RMSD) for the protein’s specific components (Cα, backbone, side chain, and heavy atoms) as well as the ligand fitted onto the protein, derived from 100 ns simulation trajectories. The docked pairing of telomerase and TQ revealed tolerable fluctuations in the Cα atoms, maintaining equilibrium throughout the duration of the simulation, as depicted in Figure 7A. Similarly, the backbone, side chain, and heavy atoms of telomerase extracted from the corresponding complexes demonstrated consistent deviations under 3 Å. Furthermore, the RMSD of TQ adjusted to the protein displayed acceptable variations. Notably, the TQ–protein complex exhibited the most significant deviation, yet it remained within an acceptable range of under 8 Å by the conclusion of the 100 ns simulation. These data suggested that the TQ-docked telomerase complexes achieved a state of equilibrium and stability over the course of the 100 ns simulation. Two hydrogen bonds were formed on average over the simulation time (Figure 7B). The overall RMSF was also computed for the telomerase proteins. Between residues 270 and 290, the protein showed higher fluctuations than all other residues (Figure 7C). Furthermore, the radius of gyration, as depicted in Figure 7D, indicated a stable distribution of protein atoms around its central axis, with measurements fluctuating between 24 and 25 Å. This provided further evidence of the complex’s stability and equilibrium throughout the simulation.

## 4. Discussion

We propose that utilizing SBE-β-CD to enhance the solubility and stability characteristics of TQ holds great potential as a treatment strategy for leukemia, a condition characterized by the abnormal proliferation of blood cells. SBE-β-CD is an attractive choice due to its low hemolytic activity compared to other cyclodextrin derivatives [16]. This makes it a safe option for encapsulating TQ by SBE-β-CD as an anti-leukemic therapy. Additionally, SBE-β-CD has been successfully investigated for controlled drug release and antibacterial efficacy using nanofibers and nanoparticles techniques [17]. Previously, we investigated the formation of inclusion complexes between TQ and SBE-β-CD to improve the solubility and delivery of TQ, and the results demonstrated that the inclusion complexes significantly enhanced TQ’s solubility and exhibited improved anticancer effects, suggesting the potential of SBE-β-CD as a drug delivery system for TQ, though further research is needed [13]. Anticancer therapy focuses on targeting telomeres and telomerase, as they play crucial roles in cancer development. Telomerase, an enzyme responsible for regulating telomere length, is activated in nearly all cancer cells, allowing for uncontrolled cell growth. Consequently, telomerase has become a prominent target in cancer treatment research. Natural products that deactivate telomerase and destabilize telomeres present promising opportunities for the development of new targets in cancer therapy [18]. According to one study, TQ promoted telomerase attrition by preventing the telomerase enzyme from functioning in human glioblastoma cancer cells [19]. However, there are limitations to using TQ in drug development, as TQ is a poorly water soluble bioactive compound, which shows poor bioavailability [20]. Therefore, in this study, we encapsulated TQ with a type of cyclodextrin (CD) known as SBE-ß-CDs to enhance the bioavailability of TQ [21]. CDs are employed as drug carriers in the pharmaceutical industry to improve the solubility, stability, and bioavailability of bioactive substances. CDs are friendly to humans because of their high level of compatibility and are FDA (Food and Drug Administration) approved [22]. 

The aim of this study was to assess the effect of telomerase downregulation on suppressing the growth of leukemia cells by conducting an annexin V-FITC test. This test was performed to investigate whether the reduced telomerase activity in leukemia cells resulted in apoptosis, a process in which phosphatidylserine (PS) translocates to the outer plasma membrane. By studying PS translocation, apoptosis could be examined as a potential mechanism.

The cytotoxic effect of TQ and TQ/SBE-ß-CDs were tested on K-562 cells. Even though the cytotoxic results showed that TQ and TQ/SBE-ß-CDs had a low toxic effect after an incubation time of less than 72 h, the complex of TQ/SBE-ß-CDs showed better toxicity effects at lower concentrations than TQ alone, which could have been due to the enhanced solubility properties of the complex. It was noticed as well that for incubation times of less than 72 h, the cytotoxicity effects reached a plateau, even at low concentrations, indicating that TQ and TQ/SBE-ß-CDs presented an inhibition effect and prevented cell division or interfered with the metabolism process, as the CCK8 assay was affected by the enzymatic activity of the cells. However, the plateau effect did not show up at a low concentration after 96 h of treatment, and at the same time the cytotoxic effects increased by over 50%, indicating that the TQ and TQ/SBE-ß-CDs cytotoxicity was time-dependent, not concentration-dependent. However, the shift in activity that happened after 96 h could have resulted from the activity capacity of TQ and TQ/SBE-ß-CDs, as the number of cells increased and all the available substrate had already interacted with the target enzymes, which justified the loss in toxic activity, especially at low concentrations, and the increase in the activity at high concentrations. However, our results could be specific to the nature of K-562 cells and the structure and purity of TQ that has been isolated from black seed. A previous study explored the cytotoxic effects of TQ on primary hepatocytes and showed a significate necrosis effect of TQ at a concentration of 3.28 μg/mL, and concentrations higher than 3.28 μg/mL caused acute cytotoxicity within hours after treatment [23]. The anticancer properties of TQ have been extensively investigated in various cancer types, and certain effects seem to be linked to the regulation of key genes involved in cancer biology. In the case of breast cancer, for example, TQ has been observed to increase the expression of p53, a vital tumor-suppressor gene, in a time-dependent manner. This upregulation of p53 promotes apoptosis and inhibits the proliferation of cancer cells [24]. In acute myeloid leukemia cells, TQ has been shown to re-express tumor-suppressor genes through de-methylation, inhibit the enzymatic activity of JAK/STAT signaling, and induce apoptosis [8]. In our previous study, it was observed that the combination of TQ with SBE-β-CD, namely TQ/SBE-β-CD, exhibited superior inhibitory effects on breast and colon cancer cell lines compared to TQ alone. Notably, the strongest inhibitory effects were observed on SkBr3 and HT29 cell lines, which are characterized by being estrogen- and progesterone-independent and possessing unique genetic and molecular characteristics. TQ exerts its effects by modulating oncogenic pathways, suppressing inflammation and oxidative stress, inhibiting angiogenesis and metastasis, and inducing apoptosis. However, the clinical application of TQ is hindered by its limited solubility and absorption. By utilizing a nanoformulation approach such as SBE-β-CD, the effectiveness of TQ can be significantly enhanced. Thus, TQ/SBE-β-CD holds great promise as a potential strategy for cancer treatment [13].

Annexin V is a protein that binds to phospholipids and requires calcium, showing a strong attraction to PS. It is commonly utilized alongside PI, a fluorescent dye, to distinguish between apoptotic and necrotic cells. [25]. In order to quantify apoptotic leukemia cells after treatment with TQ and its encapsulated version with SBE-ß-CDs, cells were exposed to annexin V/PI staining and analyzed using flow cytometry. The results indicated that the cells treated with SBE-ß-CDs exhibited the highest percentage of viable cells (95.8%), with negligible percentages of apoptotic, dead, and necrotic cells (Figure 1), thus confirming the safety of the encapsulation method employed in this study. Additionally, the TQ/SBE-ß-CDs complex increased the percentage of late apoptosis by 6% compared to TQ treatment alone. Moreover, the percentage of necrotic cells in the TQ/SBE-ß-CD-treated cells was elevated by 15.7%. This increase in apoptosis may be attributed to the downregulation of telomerase, accompanied by DNA fragmentation and the activation of apoptotic genes [26], which requires further investigation in the future. This finding would imply that TQ/SBE-ß-CDs can cause necrosis, a type of cell suicide that leads to more inflammation and may not be the preferred approach for cancer therapy. Necrosis is a type of cell death that occurs in addition to apoptosis. Here, we recommend additional research to improve the apoptotic effect of TQ/SBE-ß-CDs while minimizing necrosis induction.

In the subsequent phase, our aim was to comprehend the activation of the caspase cascade during SBE-ß-CD, TQ, and TQ/SBE-ß-CD-induced apoptosis in leukemia cells. To achieve this, we examined the activities of caspases 3/7, 8, and 9 after treating the cells with varying concentrations of each treatment for 96 h. As depicted in the figure, the combined treatment led to the activation of caspase 9, along with a concentration-dependent activation of caspase 8 (at concentrations ≥12.5 µg/mL). This observation suggested the involvement of both extrinsic and intrinsic apoptosis pathways in the inhibition of cancer cells through TQ/SBE-ß-CD treatment. Another study investigated the impact of thymoquinone on human glioblastoma cells and normal cells, revealing that glioblastoma cells exhibited higher sensitivity to thymoquinone-induced antiproliferative effects. Thymoquinone induced DNA damage, cell cycle arrest, and apoptosis in the glioblastoma cells. Moreover, thymoquinone facilitated telomere attrition by inhibiting the activity of telomerase. These researchers also explored the role of DNA-PKcs in the changes in telomere length mediated by thymoquinone, and they discovered that glioblastoma cells with DNA-PKcs were more responsive to thymoquinone-induced effects compared to cells lacking DNA-PKcs [19]. Thus, this study suggests that this complex treatment may induce apoptosis through different apoptosis pathways. 

Telomerase is a significant enzyme involved in the preservation of chromosome ends, thereby playing a crucial role in cell division and aging. It becomes particularly relevant in the context of cancer research, as cancer cells often exhibit high telomerase activity, which contributes to their unchecked growth and division [27]. Our findings revealed that the telomerase activity demonstrated a quantitative fold reduction among the cells treated with SBE-ß-CDs, TQ, and TQ/SBE-ß-CDs as compared to the negative control (untreated cells). On closer examination, it was found that the effects exhibited by SBE-ß-CDs were almost similar to those of the negative control, signifying that the impact of SBE-ß-CDs on telomerase activity was minimal. On the other hand, the complex of TQ/SBE-ß-CDs showed a significantly higher impact, effectively reducing the telomerase activity by more than double when compared to TQ alone. This suggested that the TQ/SBE-ß-CD complex could potentially have stronger anti-telomerase properties than TQ in isolation, which is believed to be related to the increase solubility of TQ when combined with SBE-ß-CDs. It is worth noting that previous studies have already established that the exposure of thymoquinone to telomerase-positive human foreskin fibroblast cells (hTERT-BJ1) results in a significant reduction in telomerase activity and causes notable telomere attrition, which reinforces the anti-telomerase potential of TQ [19].

Molecular docking is a key tool in computational molecular biology. It is used to predict the interaction of a small molecule (ligand) with a protein (receptor) at the molecular level, which can provide important insights into the behavior of biomolecules. The interaction between the SBE-ß-CDs and thymoquinone complex was explored using a molecular docking approach that showed two hydrogen bonds playing a crucial role in the stability of the SBE-ß-CD and thymoquinone complex, suggesting that these specific residues might be important for the potential of SBE-ß-CDs to bind with thymoquinone. Furthermore, the results of the study showed that TQ interacted with chain A of the telomerase. The top-scoring pose (mode 1) had a binding affinity of −6.65 kcal/mol, corresponding to a Ki of 13.45 μmol. This suggested a strong binding affinity between telomerase and thymoquinone. The binding affinity is a measure of the strength of the interaction between the ligand and the receptor. A lower value (more negative) indicates a stronger interaction. The second and third highest scoring poses had a low lower bound of the RMSD (1.97 and 1.55 Å, respectively) and upper bound of the RMSD (3.08 and 3.81 Å, respectively) from the ligand’s best mode (mode 1), indicating the relative stability of the interaction. The RMSD (root mean square deviation) was a measure of the average distance between the atoms of the ligand in the different poses. A lower RMSD indicated a higher similarity between the poses, suggesting that the interaction was relatively stable. The thymoquinone in our study formed H bonds with two amino acids of chain A, THR 143 and TYR 116, and hydrophobic interactions with several amino acids of chain A, including ALA 81, LEU 95, ALA 81, TYR 84, TYR 118, TYR 123, and TRP 147, which originated from a different binding site to that targeted by 1,3,4-oxadiazole derivatives containing 1,4-benzodioxan moieties as potential telomerase inhibitors by binding to the telomerase and forming H bonds with two amino acids, LYS 372 and LYS 406 [28]. 1-isoquinoline-2-styryl-5-nitroimidazole derivatives were proposed to act as telomerase inhibitors by forming H bonds with two amino acids, LYS 189 and ALA 255 [29].

Our findings demonstrated that TQ, particularly when complexed with SBE-β-CD, induced a significant reduction in telomerase activity in leukemia cells. This was evidenced by the decreased levels of hTERT mRNA, the primary determinant of telomerase activity, and the decrease in telomere length, as observed in the fluorescence in situ hybridization (FISH) analyses. While these results are encouraging, we must acknowledge the limitations of our current understanding of the mechanisms through which TQ/SBE-β-CD achieves these effects. Firstly, while our results indicated that TQ/SBE-β-CD reduced telomerase activity, we did not identify the precise mechanisms involved. TQ/SBE-β-CD could directly interact with telomerase or it could indirectly affect telomerase activity by modulating the associated signaling pathways or cellular processes. For instance, TQ could interfere with the JAK/STAT pathway, which has been reported to be essential for the maintenance of telomerase activity in certain cancer cells. Further investigation into these underlying mechanisms is required. Secondly, while we observed a reduction in telomere length following TQ/SBE-β-CD treatment, our current study did not clarify whether this was due to the inhibition of telomerase or the increased telomere erosion through the induction of DNA damage. Given the cytotoxic effect of TQ/SBE-β-CD complex on K-562 cells, there is a possibility that the observed telomere attrition may have been partially due to DNA damage. Future studies incorporating DNA damage assays could provide clarity in this regard. Thirdly, the cytotoxicity of the TQ/SBE-β-CD complex was assessed primarily on the K-562 cell line. While these findings provide significant insights, the response of cancer cells to treatment can be highly variable depending on their genetic and molecular background. Therefore, the outcomes we observed in this study may not be applicable to other leukemia subtypes or other cancer types. Further studies encompassing a wider array of cancer cell lines and including a comparison with normal cells are necessary to validate the generalizability of our findings. Therefore, understanding how to manipulate TQ/SBE-β-CD to favor apoptosis over necrosis could significantly enhance its therapeutic potential. Lastly, in vivo studies are crucial as the next step in advancing the functionality of TQ/SBE-β-CD. Therefore, further investigations are warranted utilizing an appropriate animal model. Despite these limitations, our study provided preliminary evidence supporting the potential of TQ/SBE-β-CD as an anti-leukemic therapy. However, more comprehensive studies investigating the exact mechanisms of action, the role of TQ/SBE-β-CD in DNA damage, and its effect on a wider range of cell lines and using animal models are required before TQ/SBE-β-CD can be considered a viable anti-leukemic therapeutic. It is also important to explore strategies that could enhance the apoptotic effects of TQ/SBE-β-CD while minimizing necrosis.

## 5. Conclusions

In conclusion, this study demonstrated the potential of utilizing SBE-β-CD to enhance the solubility and stability of thymoquinone (TQ) for the treatment of leukemia. The encapsulation of TQ with SBE-β-CD resulted in improved solubility, enhanced cytotoxic effects, and increased apoptotic activity, making it a promising drug delivery system for TQ. The TQ/SBE-β-CD complex exhibited heightened apoptotic effects, involving both intrinsic and extrinsic pathways, while effectively reducing telomerase activity compared to TQ alone. Molecular docking studies further supported the strong binding affinity between TQ and telomerase, indicating its potential as an anti-telomerase agent. Quantitative results revealed significant cytotoxicity, with maximum inhibition effects of 69.7 ± 5.72% for TQ and 89.1 ± 1.31% for TQ/SBE-β-CDs after 96 h of treatment. The half-maximal inhibitory concentrations (IC50) for TQ and TQ/SBE-β-CDs were determined to be 8.5 µg/mL and 9.9 µg/mL, respectively. Moreover, the telomerase activity of TQ and TQ/SBE-β-CDs was significantly reduced by 18.70 ± 2.199 fold and 43.46 ± 5.568 fold, respectively, compared to the negative control. While the findings present promising avenues for leukemia treatment, further research is needed to optimize apoptotic effects while minimizing necrosis induction. Additionally, rigorous investigations and safety studies are essential to validate the clinical application and safety profile of SBE-β-CD as a drug delivery system. In summary, this study supports the potential of TQ/SBE-β-CD as a promising strategy to target telomerase and enhance the cytotoxic effects of TQ in leukemia treatment.

## Figures and Tables

**Figure 1 biomedicines-11-01891-f001:**
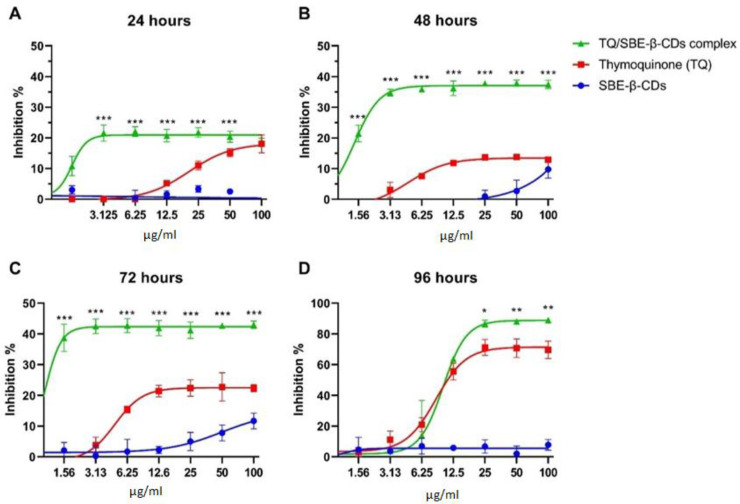
(**A**–**D**) Cytotoxicity results of leukaemia cells (K-562) after 24, 48, 72, and 96 h of the treatment. The control group was untreated cells. Data are presented as mean ± SD. * *p* < 0.05, ** *p* < 0.01, *** *p* < 0.001 indicate significant differences between TQ and TQ/SBE-ß-CDs for the same concentration level.

**Figure 2 biomedicines-11-01891-f002:**
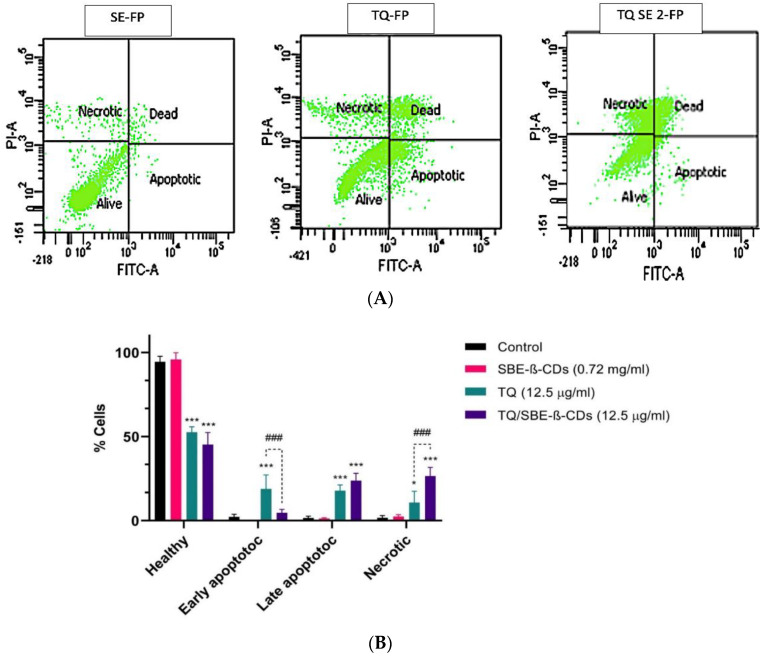
(**A**) Flow cytometry graphs and result quantification in leukaemia cells after 96 h of treatment. (**B**) The control group remained untreated. Data are presented as mean ± SD; * *p* < 0.05, *** *p*< 0.001 indicate significant differences compared to control; ### *p* < 0.001 indicates significant differences between TQ- and TQ/SBE-ß-CDs-treated groups.

**Figure 3 biomedicines-11-01891-f003:**
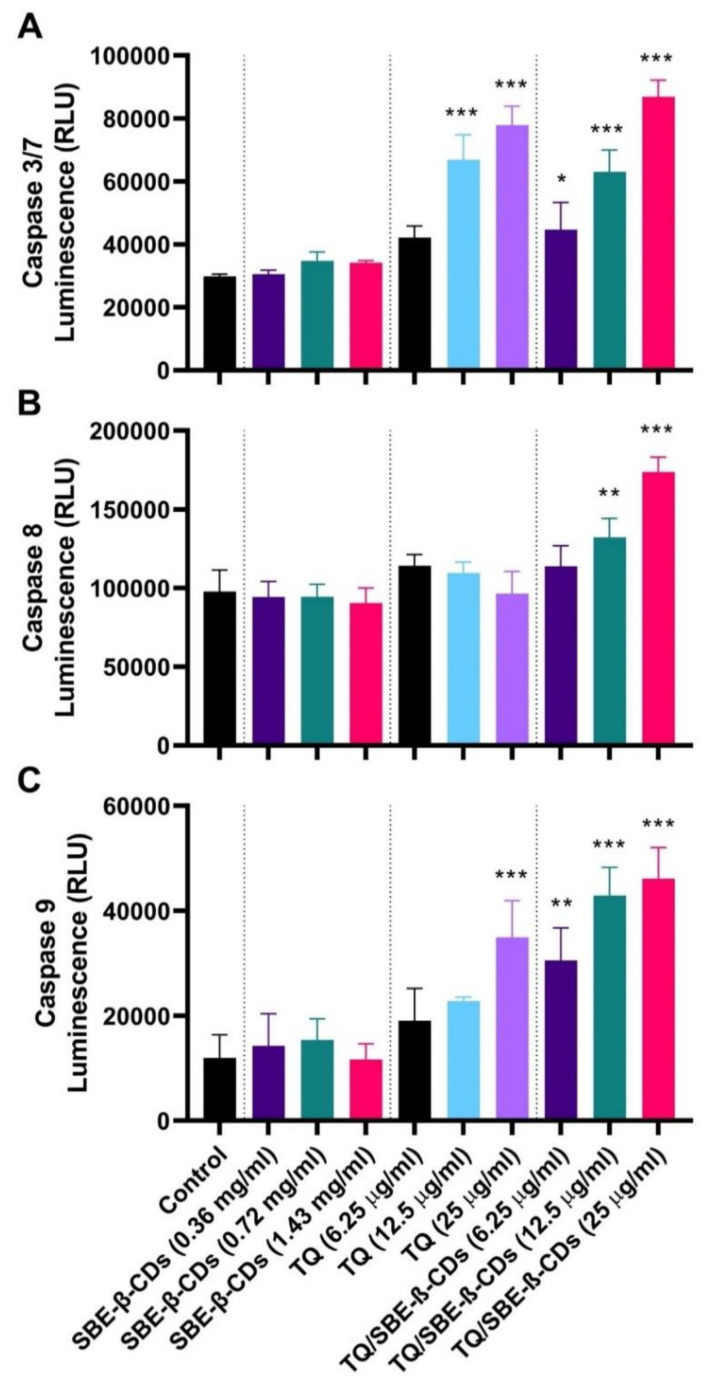
(**A**) Caspase 3/7, (**B**) Caspase 8, and (**C**) Caspase 9 activities induced by TQ, SBE-ß-CDs, and TQ/SBE-ß-CDs therapy in leukemia cells. Estimation was carried out using luminescence analysis at 96 h treatment. Data are presented as mean ± SD. * *p* < 0.05, ** *p*< 0.01, *** *p* < 0.001 indicates a significant difference compared to the control cells (untreated).

**Figure 4 biomedicines-11-01891-f004:**
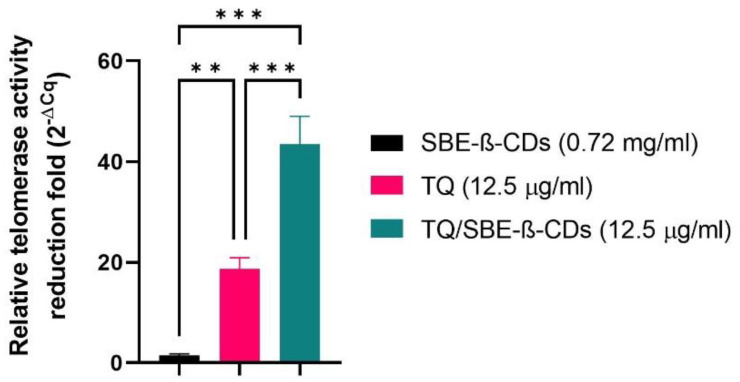
Quantitative PCR results of telomerase activity. Numbers represent the average fold reduction compared to the untreated cells. ** *p* < 0.01, *** *p* < 0.001.

**Figure 5 biomedicines-11-01891-f005:**
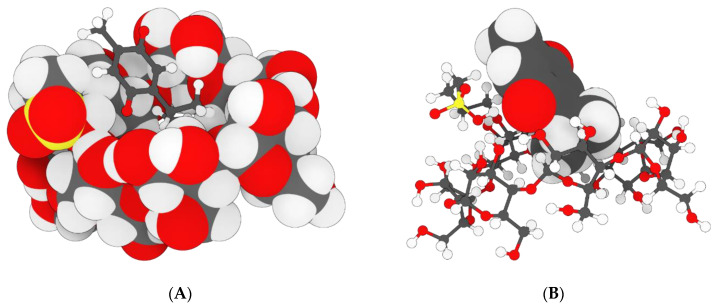
Molecular docking modeling of thymoquinone with SBE-ß-CDs. (**A**) interaction geometry of TQ with SBE-β-CD; (**B**) interaction geometry of TQ with SBE-β-CD; (**C**) the interaction geometry of TQ with SBE-β-CD; (**D**) Thymoquinone.

**Figure 6 biomedicines-11-01891-f006:**
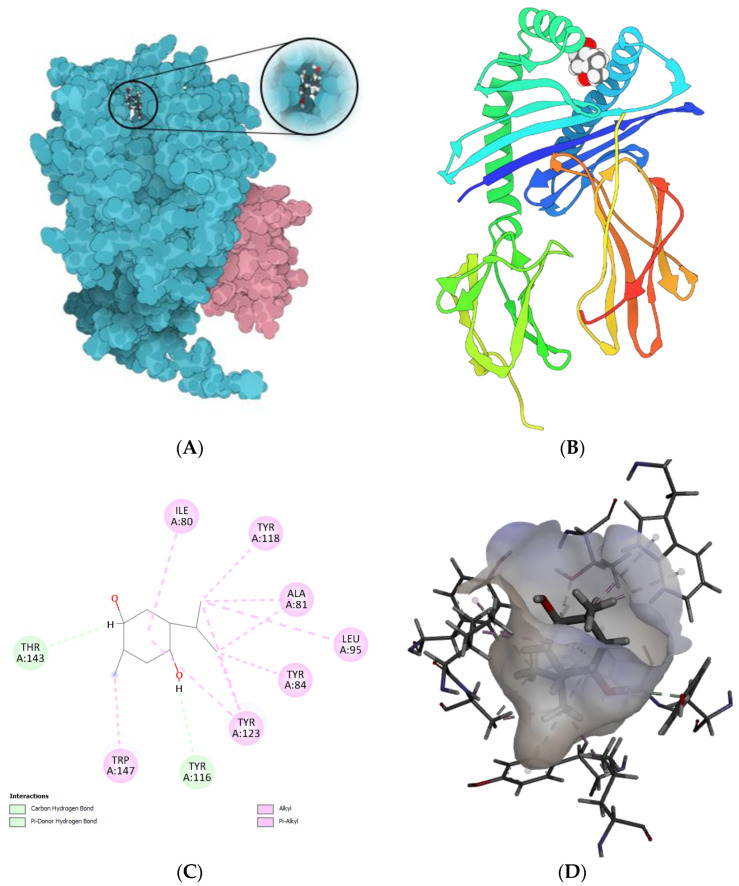
Molecular docking modeling of thymoquinone with telomerase. (**A**) telomerase; (**B**) telomerase; (**C**) Thymoquinone; (**D**) the geometric interaction of thymoquinone with telomerase.

**Figure 7 biomedicines-11-01891-f007:**
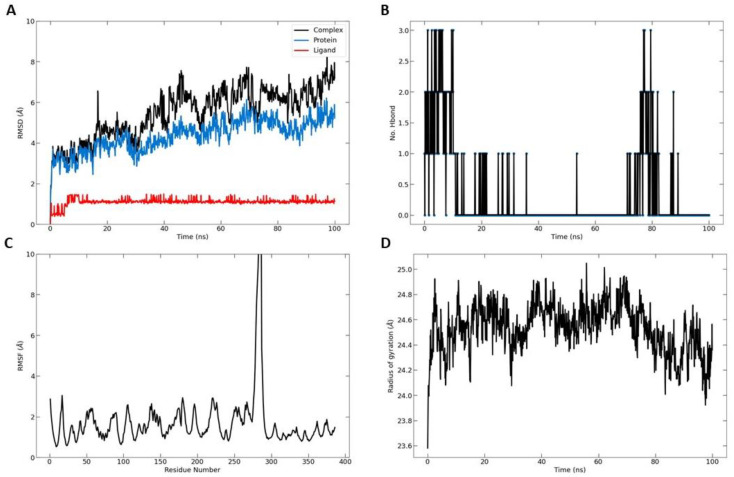
Molecular dynamic simulation results. (**A**) Root mean square deviation (RMSD) values for α-carbon atoms (blue curves) of telomerase and TQ (red curves); (**B**) H-bond number formed over the simulation time; (**C**) root mean square fluctuation (RMSF) results; (**D**) radius of gyration, plotted with respect to 100 ns MD simulation time.

**Table 1 biomedicines-11-01891-t001:** The atoms and types of the binding interactions between SBE-ß-CDs and thymoquinone.

Category	Types	Distance	From	To
Hydrogen bond	Conventional hydrogen bond	2.86	SBE-ß-CDs: H35 (H donor)	TQ: O (H acceptor)
Hydrogen bond	Conventional hydrogen bond	2.21	SBE-ß-CDs: H77 (H donor)	TQ: O (H acceptor)

**Table 2 biomedicines-11-01891-t002:** The residue and type of binding interactions between telomerase and thymoquinone.

Category	Types	Distance	From	To
Hydrogen bond	Carbon hydrogen bond	2.75	TQ: H (H donor)	Chain A: THR143:OG1 (H acceptor)
Hydrogen bond	Pi-donor hydrogen bond	2.95	TQ: H (H donor)	Chain A:TYR116 (pi-orbitals)
Hydrophobic	Alkyl	3.72	TQ: C10 (alkyl)	Chain A: ALA81 (alkyl)
Hydrophobic	Alkyl	4.67	TQ: C10 (alkyl)	Chain A: LEU95 (alkyl)
Hydrophobic	Alkyl	3.38	TQ: C9 (alkyl)	Chain A: ALA81 (alkyl)
Hydrophobic	Pi-alkyl	4.79	Chain A: TYR84 (pi-orbitals)	TQ: C9 (alkyl)
Hydrophobic	Pi-alkyl	4.38	Chain A: TYR118 (pi-orbitals)	TQ: C10 (alkyl)
Hydrophobic	Pi-alkyl	4.02	Chain A: TYR123 (pi-orbitals)	TQ: C10 (alkyl)
Hydrophobic	Pi-alkyl	3.98	Chain A: TYR123 (pi-orbitals)	TQ: C9 (alkyl)
Hydrophobic	Pi-alkyl	4.82	Chain A: TRP147 (pi-orbitals)	TQ: C7 (alkyl)
Hydrophobic	Pi-alkyl	4.97	Chain A: TRP147 (pi-orbitals)	TQ: C7 (alkyl)

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
