# Peer review of "Enhancing the Anti-Leukemic Potential of Thymoquinone/Sulfobutylether-β-cyclodextrin (SBE-β-CD) Inclusion Complexes"

_biomedicines, 2023, doi:10.3390/biomedicines11071891_

Round 1

Reviewer 1 Report

The article  'Enhancing the Anti-Leukemic Potential of Thymoquinone-SBE-β-CD Inclusion Complex' by Eid et al. explores the potential use of thymoquinone (TQ), derived from black seed, for treating leukemia. TQ has shown promising anticancer properties by inhibiting telomerase and inducing apoptosis; however, its poor solubility and limited availability in the body hinder its clinical application. To address this, the study investigates the use of SBE-β-CD, a cyclodextrin derivative, to enhance TQ's solubility. The researchers form inclusion complexes between TQ and SBE-β-CD and evaluate their effects on leukemia cells and telomerase activity. The results demonstrate improved solubility and enhanced cytotoxic effects against leukemia cells with the TQ/SBE-β-CD complex, suggesting SBE-β-CD as a potential drug delivery system. Additional assays support the anti-leukemic potential of TQ/SBE-β-CD by revealing increased apoptosis and reduced telomerase activity. Molecular docking analysis shows a strong binding affinity between TQ and telomerase. Further research is needed to optimize apoptotic effects and minimize necrosis induction. Overall, the TQ/SBE-β-CD complex offers a promising strategy for leukemia treatment, overcoming TQ's solubility and bioavailability limitations by inhibiting telomerase and enhancing TQ's cytotoxic effects.

The article is interesting and well written and could be published in Biomedicines after minor revision, since fits the scope of the journal.

The information presented in the introduction appears to be scattered and lacks a clear structure. It would be beneficial to organize the content in a logical manner, such as discussing the previous studies on thymoquinone's anticancer effects before diving into telomerase and leukemia.

Although references are provided, they are not cited properly within the text. Each statement regarding the effects of thymoquinone should be supported by specific references to strengthen the credibility of the claims made.

While the introduction briefly mentions a previous study on acute myeloid leukemia and thymoquinone's effects on tumor suppressor genes, it does not clearly outline the novel contribution or research gap that the current study aims to address. The introduction should emphasize the unique aspects or contributions of the research to distinguish it from previous work.

The description of the lyophilization technique and the replication of a prior process are insufficiently detailed. It would be helpful to provide more specific information on the lyophilization process, such as the freeze-drying parameters, the composition of the cryoprotectant, and the duration of the lyophilization process.

The purpose of using a balanced 1:1 molar combination of TQ and SBE-β-CD is not clearly stated. The rationale behind this choice should be explained.

The description of cell culture conditions lacks essential details. It would be important to specify the number of plates used, the number of wells per plate, and the volume of the medium in each well.

The description of the assays is unclear and contains typographical errors. It is important to provide a more detailed explanation of the assay procedure, including the specific steps followed and the purpose of each assay. Also, correct typographical errors such as "kites" to "kits."

The meaning and calculation of ΔCq and relative telomerase reduction activity should be explained in more detail.

The description of the molecular docking studies is lacking details. It would be beneficial to mention the specific software settings and parameters used for the AutoDock VINA docking, such as the search algorithm, scoring function, and any applied constraints.

The description of the molecular dynamics simulation is insufficient. More details should be provided regarding the simulation setup, such as the force field parameters used, the integration time step, and the simulation protocol (equilibration, production run). Additionally, the choice of a 100 ns simulation time should be justified.

The Results mentions significant differences between groups without providing specific statistical tests used or the actual p-values. It is important to include the statistical analysis performed and the significance level to support the claims of significance.

The section presents some results as percentages, some as fold reductions, and some as IC50 values. It would be clearer and more consistent to present all results in a consistent format (e.g., percentages or fold changes) to facilitate comparisons and interpretation.

The discussion lacks critical analysis of the results and their interpretation. The authors mainly present the findings without considering alternative explanations or limitations of the study. A more balanced discussion that acknowledges potential weaknesses or alternative interpretations would strengthen the section.

The Conclusions paragraph does not provide specific quantitative results or data to support the claims made. Instead, it uses vague language such as "promising potential," "proven success," and "enhanced apoptotic effects" without providing concrete evidence or statistics. The statement that SBE-β-CD "serves as a safe" option lacks supporting evidence or reference to safety studies. It is important to provide data or references to substantiate claims related to safety, especially when discussing potential treatment strategies.

The following criticism addresses formal issues: Figures 7A and 7D are referenced in the text, but Figures 7B and 7C are not mentioned. Additionally, in line 388, Figure 3 should be referenced.

The sentences are well-structured, and there is a clear flow of ideas. The vocabulary and terminology used are appropriate for the subject matter, and technical terms are used accurately. There are no apparent issues with spelling, punctuation, or grammar. Only those already mentioned in the Suggestions to Authors.

Author Response

The file is attached 

Reviewer 2 Report

Reviewer’s comments

The authors investigated the anti-leukemic potential of TQ/SBE-β-CD complex, an inclusion complex of TQ and SBE-β-CD, in this manuscript. Results showed that the solubility and cytotoxic effect of TQ/SBE-β-C were enhanced compared to TQ alone. In addition, leukemia cells treated with TQ/SBE-β-CD showed increased apoptosis and reduced telomerase activity. Furthermore, molecular docking analysis revealed a strong binding affinity between TQ and telomerase.

The manuscript is very well organized with detailed analysis of the enhanced anti-leukemic potential of TQ/SBE-β-CD, and I think this manuscript will provide useful information to the readers of Biomedicines. However, the minor revisions are necessary for acceptance as follows:

1. From the viewpoint of safety of TQ/SBE-β-CD, it is important to compare its cytotoxic effect on K-562 cells with that on noncancer cells (normal cells). Please comment in the discussion if you have data on the cytotoxic effects on normal cells.

2. In vivo studies are important as the next step to develop the functionality of TQ/SBE-β-CD. Please comment in the discussion on the prospects for in vivo studies using experimental animals and clinical studies on human leukemia.

Author Response

The reply file is attached 
